# Study on Improvement of Radio Propagation Characteristics of Cast Iron Boxes for Water Smart Meters

**DOI:** 10.3390/s23249716

**Published:** 2023-12-08

**Authors:** Eiichi Tateishi, Yuantong Yi, Nobuhiro Kai, Takaya Kumagae, Tatsuya Yamaguchi, Haruichi Kanaya

**Affiliations:** 1Graduate School of Information Science and Electrical Engineering, Kyushu University, Fukuoka 819-0385, Japan; yi.yuantong.646@s.kyushu-u.ac.jp (Y.Y.); kanaya@ed.kyushu-u.ac.jp (H.K.); 2Hinode Holdings Ltd., Fukuoka 812-8636, Japan; n-kai@hinodesuido.co.jp (N.K.); t-kumagae@hinodesuido.co.jp (T.K.); 3HINODE, Ltd., Fukuoka 812-8636, Japan; t-yamaguchi@hinodesuido.co.jp

**Keywords:** IoT, smart meter, cast iron, radio wave, LPWA (low-power wide-area)

## Abstract

Water utilities in Japan face a number of challenges, including declining water demand due to a shrinking population, shrinking workforce, and aging water supply facilities. Widespread use of smart water meters is crucial for solving these problems. The widespread use of smart water meters is expected to bring many benefits such as reduced labor by automating meter reading, early identification of leaks, and visualization of pipeline data to strengthen the infrastructure of water services, business continuity, and customer service, as detailed data can be obtained using wireless communication. Demonstration tests are actively conducted in Japan; however, many problems have been reported with cast iron meter boxes blocking radio waves. To address the issue, a low-cost slit structure for cast iron meter boxes is investigated in this study. The results confirm that the L-shaped tapered slit array structure with a cavity, which can be fabricated in a cast iron integral structure, satisfies the design loads required for road installation. The proposed slit structure achieved gain characteristics from −3.32 to more than 9.54 dBi in the 800 to 920 MHz band. The gain characteristics of conventional cast iron meter boxes range from −15 to −20 dBi, and the gain has been significantly improved. Antennas with a gain of −2.0 to +1.5 dB (0.8 to 2.5 GHz) were used for the transmitter antenna, which was found to have a higher gain than the transmit antenna in the 800 to 880 MHz frequency band. In the 1.5 to 2.0 GHz band, a high peak gain of 4.25 dBi was achieved at 1660 MHz, with no null and the lowest gain confirmed that this is an improvement of more than 10 dBi over conventional products.

## 1. Introduction

Japan’s water utilities are facing challenges such as a declining population, aging facilities, and a decrease in the number of water utility employees. Therefore, it is necessary to optimize and improve the efficiency of water services to sustain healthy and safe water services. Water meters are installed to calculate charges, and meter readers visit each household every other month or every month, spending considerable time, effort, and resources on meter readings. In addition, in areas where meter reading is difficult, such as those with heavy snowfall or remote islands, it is necessary to consider measures to ensure the safety of meter readers and countermeasures against delays in meter reading. In response to these issues, smart water meters using wireless communication not only save labor in meter reading, but also enable detailed data, including demand fluctuations, to be obtained, which is expected to have many benefits, such as improving the efficiency of energy use, identifying water leakage at an early stage, and improving user services. Therefore, demonstration tests are actively being conducted to promote the use of smart water meters as a technology that can solve various problems faced by water utilities [1,2,3].

The data flow using smart water meters is shown in Figure 1. Smart water meters generally upload information from underground water meters to the cloud via wireless communication. The results of demonstration tests have reported a number of issues related to the stability of wireless communication. As the water meter is buried underground, radio waves find it difficult to penetrate, and the cast iron meter box also makes it difficult for radio waves to penetrate [4,5].

Plastic meter boxes, which allow radio waves to pass through easily, are also available, but cannot be used in areas where vehicles enter because the design load [6] is set based on the installation environment. Manhole antennas have been developed to monitor the conditions inside manholes via wireless communication [7,8]. Manhole antennas have a built-in communication antenna on the surface of the steel cover, which enables stable wireless communication and has a load-bearing capacity that allows for installation on roadways. However, manhole antennas are large and expensive, with diameters of 600 mm or greater, making their application in smart water meters difficult. Therefore, there is a need for a cast iron meter box that can be installed in a vehicular traffic environment while ensuring stable radio communication characteristics of smart water meters.

Currently, several low-power wide-area (LPWA) communication methods are used for wireless communication in smart water meters. The long-term evolution of machines (LTE-M) and Narrowband Internet of Things (NB-IoT) are commonly used communication methods. These communication methods use mobile phone networks, but consume less power. Long Range (LoRa) and SigFOX, which do not require a license, have also been widely adopted [9,10,11,12,13]. LTE-M and NB-IoT, which are representative of LPWA, use the 850 to 900 MHz band and the 1.5 to 2.0 GHz band, as described in “Communication methods and frequency bands of each mobile operator” [14] by the Ministry of Internal Affairs and Communications. LoRa and Sigfox also use the 920 MHz band.

Therefore, cast iron meter boxes for smart water meters need to be able to transmit radio waves in the 850 to 920 MHz and 1.5 to 2.0 GHz frequency bands. In addition, cast iron meter boxes cannot be designed with large openings because they are installed at vehicle access points; therefore, they need to be strong enough to prevent soil and sand from entering and high heels from sinking into the box. Furthermore, castings generally require a wall thickness of several millimeters or more and have manufacturing constraints that do not allow for complex shapes or high dimensional accuracy. On the other hand, clear standards are missing for radio transmission characteristics, and defining radio transmission is also an issue. The aim of this study is to ensure that the antenna gain characteristics of the smart water meter and do not generate nulls in the target frequency band and that the meter has characteristics to efficiently radiate radio waves to the ground.

The main challenge is to design a simple, low-cost, and feasible structure capable of transmitting radio waves in the target frequency band, while satisfying the basic requirements of cast iron meter boxes.

This paper describes the slit structure of cast iron meter boxes and explains the effect of an L-shaped slit covering the target frequency band. It then describes a method for improving radio transmission in a specific frequency band, while fulfilling the basic performance and manufacturing constraints of cast iron meter boxes. This method uses an L-shaped tapered slit array structure with cavities that can be manufactured into an integral cast iron structure. Finally, a cast iron meter box with a structurally optimized L-shaped tapered slit array structure and a cavity was fabricated and tested.

This paper is structured as follows: Section 2 provides an overview of cast iron meter boxes for smart water meters; Section 3 describes the slit structure design concept and the characteristics of the slit structure (slit #1), which is constructed on a simple cast iron flat plate, followed by a description of the characteristics of the L-shaped tapered slit array structure with a cavity (slit #2), which was optimally designed to fulfil the basic requirements of a cast iron meter box. In Section 4, the results of the prototype evaluation and comparison with the analysis results are presented. The final section (Section 5) provides the study conclusions.

## 2. Cast Iron Meter Box for Smart Water Meters Overview

Figure 2 shows an overview of the cast iron meter box used for a smart water meter. Generally, a smart water meter contains a water meter, communication equipment, and a meter box. The meter box contains a cover, frame, and bottom, and is composed of cast iron, as the material is required to withstand the design load in environments with vehicle access.

Water meter information is regularly uploaded to the cloud via wireless communication using a communication device; however, in many cases, communication is not stable because the cast iron meter box blocks radio waves. Radio waves can leak through piping openings and gaps, but meter boxes are generally buried underground, and wireless communication from underground spaces causes high radio wave attenuation. The condition of the surrounding soil and the distance to the base station are also factors that contribute to unstable communication [15,16].

Therefore, we focused on the cover facing the ground and considered that if it were possible to improve radio wave transmission through a cast iron cover, the surrounding soil would have little effect and relatively stable communication would be possible. 

## 3. Radio Transmission Slit Structure Design for Cast Iron Meter Boxes

This section first presents the results of the optimized slit geometry and layout (slit #1). Slit #2 describes the optimized slit structure based on a structural design that considers the load-bearing capacity and manufacturing conditions.

### 3.1. Slit Shape and Layout Study (Slit #1)

First, the wall thickness of the cast iron meter box was kept constant and the shape and arrangement of the slits to be placed were studied. When the slit is rectangular, the generated magnetic field depends on the relationship between the direction of the current flowing on the metal surface and that of the slit. The literature shows that the transmitted electromagnetic field is larger when the magnetic field is perpendicular to the longitudinal direction of the slit [17,18,19,20]. In other words, the characteristics of the transmitted radio waves depend on the positional relationship between the transmitting antenna and the slit. To address this problem, L-shaped slits were placed in the four corners to bring about the change in characteristics owing to the magnetic field and direction of the slits.

The electromagnetic field analysis software FEMTET (Murata Software Co., Japan), which uses the finite element method, was used for the specific design study. The analysis model and slit structure are illustrated in Figure 3 The external dimensions of the meter box made of cast iron (459 mm long × 256 mm short × 225 mm high) were the same as those of a conventional cast iron meter box, with a flat plate wall thickness of 6 mm for the frame and bottom sections and a cover wall thickness of 8 mm. The transmitting antenna was placed at the center of a cast iron meter box. The longitudinal slit pitch A = 369 mm, longitudinal slit pitch B = 168 mm, slit width w = 5 mm, dimension of slit b fixed at 70 mm, and dimension of slit a varied between 100 and 91 mm. The length of the slit L is the dimension of a + b.

The material settings for the analysis model were obtained from the FEMTET material database, with the cast iron box set to Fe (relative permittivity: 1, non-permeability: 1.00, conductivity: 1.03 × 10^7^ s/m), which is close to FCD properties, and the antenna material set to stainless steel (relative permittivity: 1, relative permeability: 1.00, conductivity: 1.37 × 10^6^ s/m).

An adaptive mesh was used to optimize the mesh at a wavelength of 900 MHz, and calculations were performed iteratively until the change in the relative value was less than 0.02, using the S-parameter as the convergence criterion.

The computation time was 4 min and 03 s per analysis, and approximately 2.7 GB of memory was used.

The frequency characteristics of S11 (reflection coefficient) of the L-shaped slits with varying slit lengths are shown in Figure 4. The analysis revealed three major resonance points at a slit length of 161 mm. The resonant frequencies were 945, 1775, and 1843 MHz. In addition, by fixing b and changing a in the slit length L, it can be confirmed that the resonance point on the high-frequency side (1500 to 2000 MHz) does not change; however, the resonance point on the low-frequency side (800 to 1000 MHz) changes.

To confirm this behavior, the current distribution around the slit at each resonance point is shown in Figure 5. In Figure 6a, at 945 MHz, a strong current distribution can be observed at the edge of the L-shaped slit, confirming the behavior of λ/2 resonance at the entire length of the slit. At 1775 MHz in Figure 6b, a high current distribution is also observed at the L-slit corner, confirming λ/2 resonance at two locations. Figure 6c shows the resonance behavior at these two locations.

Subsequently, the radiation directivity was determined. The ground-side radiation patterns of the gain at each resonant frequency are shown in Figure 6. The cut plane at the center of the long side of the meter box housing is the XZ plane, and the cut plane at the center of the short side is the YZ plane. Moreover, 945 MHz, 1775 MHz, and 1843 MHz show no significant difference in the directivity of the gain when comparing the XZ and YZ planes, confirming that the L-shaped slit reduces polarization characteristics.

### 3.2. Slit Structure Design (Slit #2)

In this section, a reinforcement structure that can be molded as an integral structure to the backside of the slit is considered to compensate for the strength reduction. Cast iron is expected to exhibit good fatigue endurance because it is relatively easy to increase the thickness of stress concentrations and form curved surfaces at the corners, which can alleviate stress concentrations. The spheroidized graphite cast iron used in this study has characteristics superior to those of ordinary cast iron in terms of both strength and elongation. Therefore, the stress generated can be reduced by controlling the deformation behavior in areas where the strength is reduced by the slit structure. According to JIS G 5502 (Japanese Industrial Standards “Gray iron castings”), the mechanical properties of FCD600 are defined as tensile strength of 600 N/mm^2^ or more, 0.2% proof stress of 370 N/mm^2^ or more, and elongation of 3% or more [21]. The chemical composition of common cast iron is composed of iron, carbon, silicon, manganese, phosphorus, and sulfur. The amount of carbon ranges from 2.1 to 6.67%, whereas the remainder varied depending on the application [22].

The added reinforcement structure can relieve the stresses that occur under load and prevent the slit from falling into the ground when stepped on by high heels. This structure makes the slit cross-section an L-shaped waveguide, which is expected to contribute to improved radiation efficiency to the ground side through reflection. Furthermore, the slit cross-section has a tapered structure that opens towards the ground side, which improves the propagation efficiency within the slit and ensures the necessary draft angle as a cast product.

The analytical model is shown in Figure 7a, and the cross-section of the reinforcement structure is shown in Figure 7b. The dimensions, except for the slit structure section, are the same as for slit #1; frame wall thickness t1 = 6 mm, reinforcing seat wall thickness t2 = 6 mm, depth from GL face to reinforcing seat t3 = 15.5 mm, slit bottom width w1 = 5 mm, slit top width w2 = 8 mm, and w3 = 6 mm.

A comparison of the frequency gain characteristics of the slit #2 and slit #1 structures is shown in Figure 8. Slit #2 has a peak gain of 9.51 dBi in the 860 MHz band on the low-frequency side, which is an improvement of 2.24 dBi compared to the peak gain of slit #1. The bandwidth characteristics also achieved a positive gain from 800 to 948 MHz, which is ideal for the low-frequency side. In the 1.5 to 2.0 GHz band, slit #2 has improved gain in all bands compared to slit #1. High gains are observed at many points; 16.05 dBi at 1601 MHz, 17.36 dBi at 1806 MHz, 11.74 dBi at 1885 MHz and 12.87 dBi at 1964 MHz, respectively. The lowest gain was −11.8 dBi at 1915 MHz, and no nulls were observed in the entire frequency band of the target.

To confirm this behavior, the current distribution around the slit in the 860 MHz band is shown in Figure 9a, and the current distribution in the slit cross-section is shown in Figure 9b. The current distribution is strong at the bottom of the slit and on the incident side, and the electromagnetic field generated by this current is reflected inside the tapered slit and in the L-shaped waveguide, which is believed to contribute to radiation in the ground.

The results of the structural analysis of load-bearing capacity are shown in Figure 10. The structural analysis software Creo Simulate (Parametric Technology Corp., Boston, MA, USA), which uses the finite element method, was used for the structural optimization. The finite element solution used the adaptive P-method and was verified by analyzing the elastic region. The material model used had a Young’s modulus of 170 GPa and Poisson’s ratio of 0.28. The loading position was in the center of the cover, the loading plate was 200 mm × 150 mm, and the design load was evenly distributed over the loading range. Stress verification is common; however, in this case, it was verified by the displacement and generated stress that the addition of a slit structure to the conventional product caused, which did not result in a loss of strength. The results showed that the maximum displacement was 0.62 mm for slit #2 and 0.76 mm for the conventional product, confirming that the strength was equal to or greater than that of the conventional product.

Cast iron has been used in the civil engineering field for sewer manhole covers and cast iron pipes for water supply and in the road construction environment for sewer manhole covers for more than three decades. To ensure long-term durability, the allowable stresses were set considering the fatigue limit of the material [23]. Figure 11 shows the analysis results of the stress concentration area owing to the addition of the slit structure. The maximum stress at the root of the slit for the design load was 195.7 Mpa, which was below the bending fatigue limit of spheroidal graphite cast iron (245 Mpa), confirming the long-term durability of the slit.

## 4. Prototype Evaluation Results

This section first describes the specimens and test conditions used for the prototype evaluation, followed by the maximum gain and radiation characteristics obtained from the evaluation using the actual equipment.

### 4.1. Assessment Overview

A photograph of the test specimen is displayed in Figure 12. A prototype of the slit #2 structure was made with spheroidized graphite cast iron FCD600 material under conditions similar to those of the actual machine.

The evaluation was carried out in an anechoic chamber, and gain and radiation characteristics were measured using a network analyzer in the frequency range of 800 to 2000 MHz [24,25]. A biconical transmitting antenna DPA-3000 (TESEQ, Reinach, Switzerland) with broadband characteristics was used for the measurements. The experimental setup and photograph of the transmitting antenna are shown in Figure 13.

### 4.2. Results of Evaluation of Maximum Gain

Figure 14 shows a comparison of the analytical and experimental results of the frequency gain and conventional products. The prototype of the slit #2 structure from 800 to 950 MHz generally showed the same trend as the analytical results, with a peak gain of 9.54 dBi at 830 MHz, although there was a slight deviation between the analytical results and peak frequency. In the bandwidth, the effect of the shifted peak results in a positive gain in the range 800 to 890 MHz. In the target range of 800 to 920 MHz, characteristics exceeding −3.32 to 9.54 dBi are obtained. As the conventional product has a gain characteristic of −15 to −20 dBi over the entire bandwidth, the gain of our proposed one has been significantly improved.

The frequency–gain characteristics at 1500 to 2000 MHz are shown in Figure 15. The experimental results show that slit #2 exhibited a higher gain over a wider frequency band than the conventional product. Specifically, slit #2 showed a maximum gain of 4.25 dBi at 1660 MHz, and the gain gradually decreased from 1660 to 1820 MHz and increased again from 1820 MHz. On the other hand, the conventional product has a lower gain, but its frequency response shows the same trend as the proposed type. However, there is a difference between the analytical and actual evaluation results. This may be caused by the fact that in the analysis, the junction surfaces of the cover frames are precisely shielded, whereas in the actual experiment, they are affected by minute gaps and coating thickness. The above experimental results show that the proposed model has no null in any of the frequency bands and can improve the gain by approximately 10 dBi compared to the conventional model.

### 4.3. Radiation Pattern Measurement Results

Figure 16 compares the experimental and analytical radiation patterns at typical frequencies of 850, 1700, and 2000 MHz for slit #2.

The analysis and experiments at 850 MHz show a similar trend, with results close to the design aims being achieved. As the frequency increases to 1700 MHz and 2000 MHz, a trend can be observed where the difference between the experimental and analytical values increases.

## 5. Conclusions

We designed a radio wave transparent structure for a cast iron meter box compatible with smart water meters. The basic requirements of a cast iron meter box were satisfied using an L-shaped tapered slit array structure with a cavity that can be manufactured in a cast iron integral structure and includes the design loads required for installation on a carriageway. In the frequency band of 800 to 880 MHz, slit #2 achieved a gain of 1.1 to 9.54 dBi, which is higher than the gain of the transmitting antenna. The slit structure enables a high gain characteristic of 9.54 dBi at 830 MHz. On the entire high-frequency side from 1.5 to 2.0 GHz, peak gain as high as 4.25 dBi at 1660 MHz had no nulls. Analysis and experiments have confirmed that even at the lowest gain, radio transparency is improved by more than 10 dB compared with conventional products.

The analytical and experimental results show similar characteristics, which also confirm that the design is on target with regard to the assumed radio propagation characteristics.

A comparison between the proposed cast iron meter box structure (slit #2) and conventional products is presented in Table 1.

The slit #2 structure is assumed to have a higher gain than the plastic meter box because the gain characteristics of the slit #2 structure exceed the gain of the antenna alone in the 800 to 880 MHz range. The gain characteristics of the manhole antenna match the specific frequency of the communication device. The cast iron meter boxes with slit #2 met the targets in terms of design load and cost.

This result indicates that the communication performance can be improved for smart water meters buried underground, even in installations where cast iron meter boxes are required, without requiring special antenna systems. Because the structure can be manufactured by integral casting without the need for a special mold structure, mass production at a low cost is possible.

Based on the above factors, the slit structure of the cast iron meter box used in this study can reduce the cost of installing, introducing, and operating a smart water meter system and contribute to the increased use of smart water meters.

In the future, we plan to conduct field verification evaluations that consider the characteristics of environmental factors in various installation environments to confirm their practicality and limitations.

## Figures and Tables

**Figure 1 sensors-23-09716-f001:**
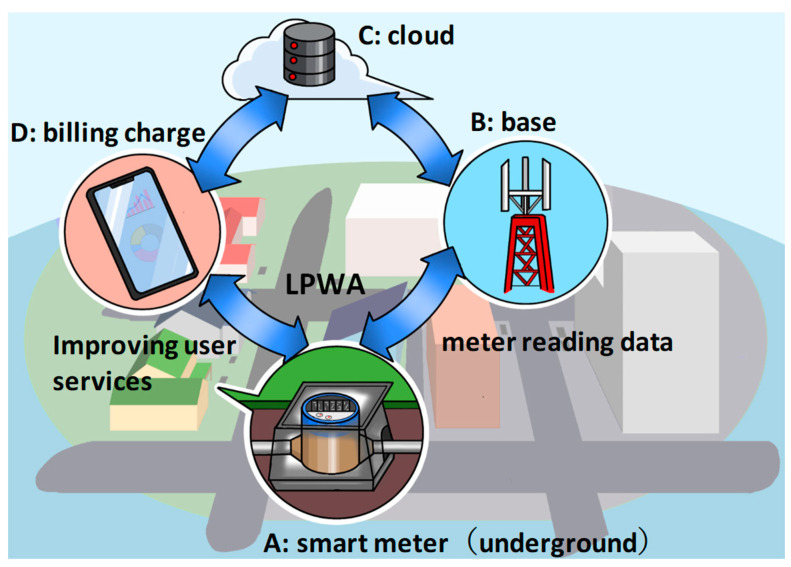
Smart water meter reading data flow.

**Figure 2 sensors-23-09716-f002:**
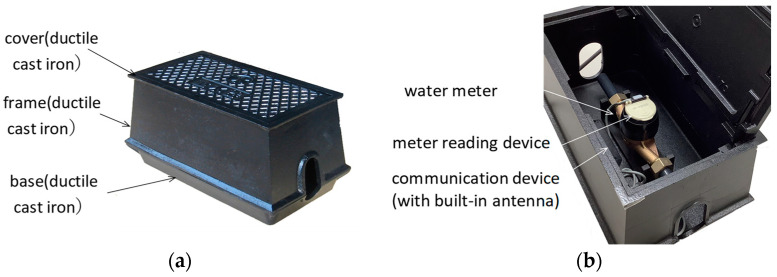
Overview of the meter box for smart water meters: (**a**) cast iron meter box exterior; (**b**) smart water meter component configuration.

**Figure 3 sensors-23-09716-f003:**
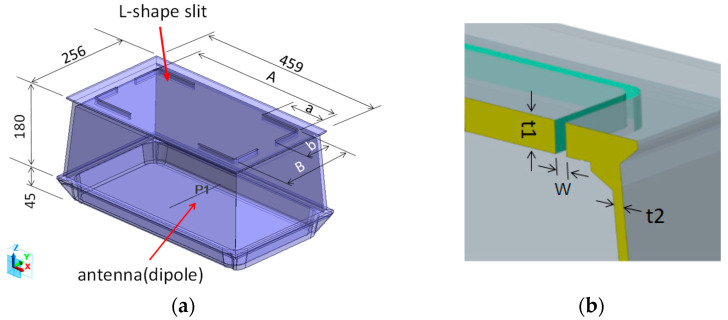
Slit #1 structural analysis model. (**a**) Isometric drawing; (**b**) cross-section view.

**Figure 4 sensors-23-09716-f004:**
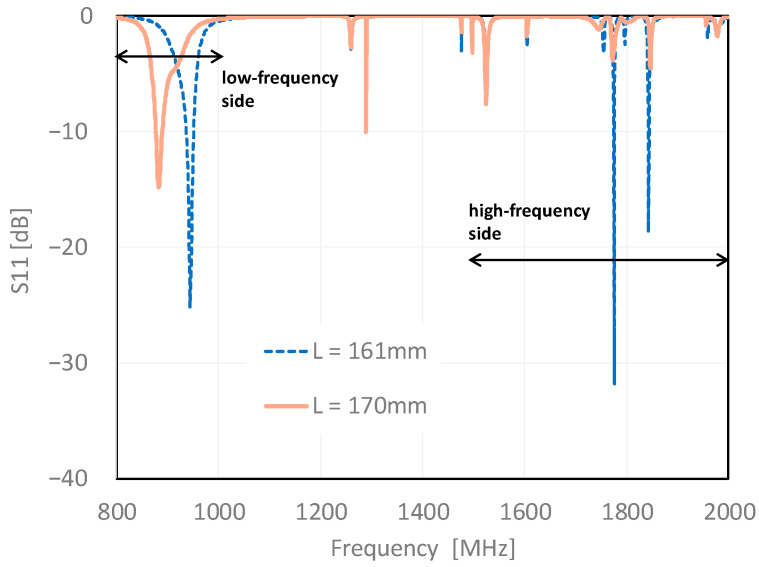
Slit #1 S11 characteristics 800 to 2000 MHz.

**Figure 5 sensors-23-09716-f005:**
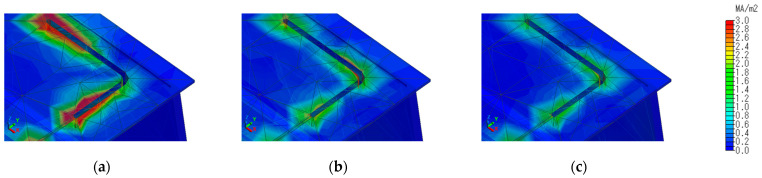
Current distribution around the slit at each resonance point. (**a**) 945 MHz; (**b**) 1775 MHz; (**c**) 1843 MHz.

**Figure 6 sensors-23-09716-f006:**
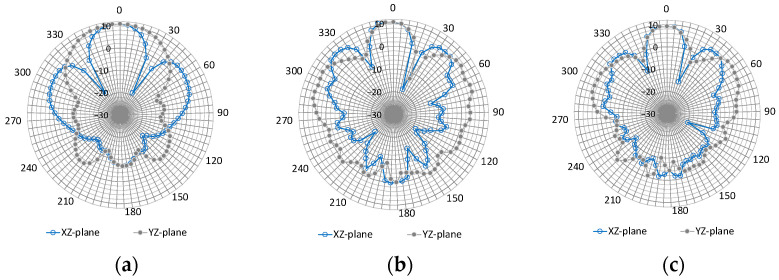
Directional gain radiation pattern. (**a**) 945 MHz; (**b**) 1775 MHz; (**c**) 1843 MHz.

**Figure 7 sensors-23-09716-f007:**
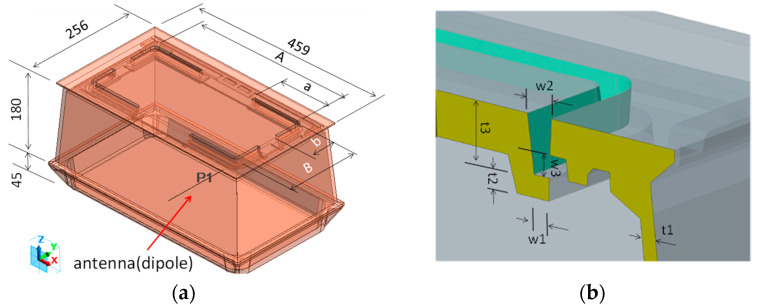
Slit #2 structural analysis model. (**a**) Isometric drawing; (**b**) cross-section view.

**Figure 8 sensors-23-09716-f008:**
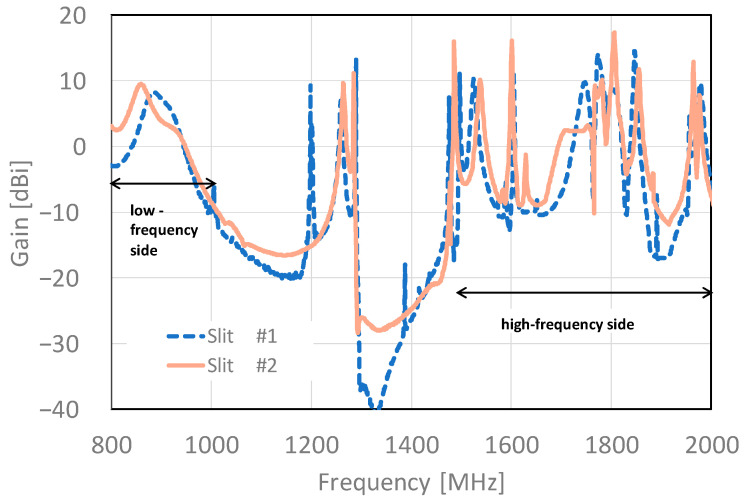
Comparison of frequency gain characteristics of slit #1 and slit #2 structures from 800 to 2000 MHz.

**Figure 9 sensors-23-09716-f009:**
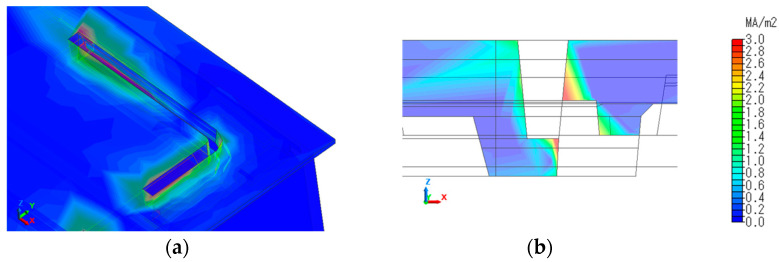
Current distribution around the slit (**a**) and across the slit (**b**) at 860 MHz for slit #2 structure.

**Figure 10 sensors-23-09716-f010:**
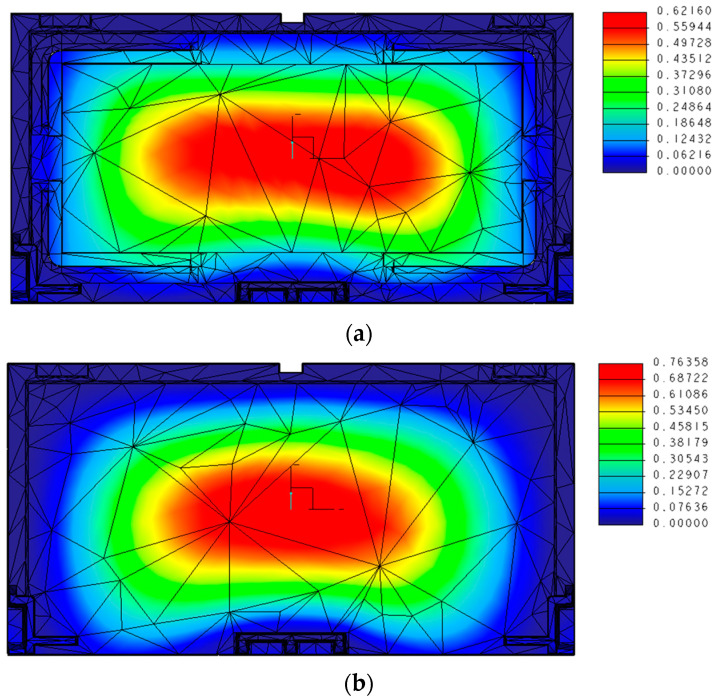
Load-bearing property analysis results (displacement). (**a**) Slit #2 and (**b**) conventional product.

**Figure 11 sensors-23-09716-f011:**
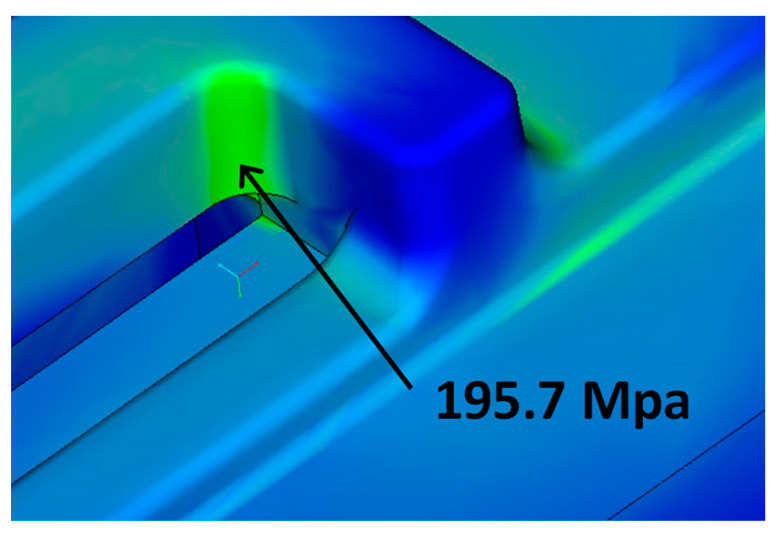
Slit structure maximum generated stress site.

**Figure 12 sensors-23-09716-f012:**
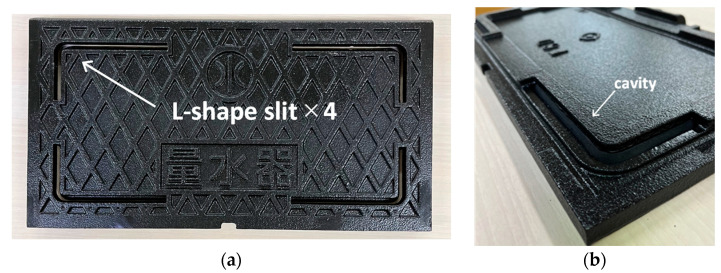
Tried and tested specimen with slit structure. Flat surface (**a**) and back surface (**b**).

**Figure 13 sensors-23-09716-f013:**
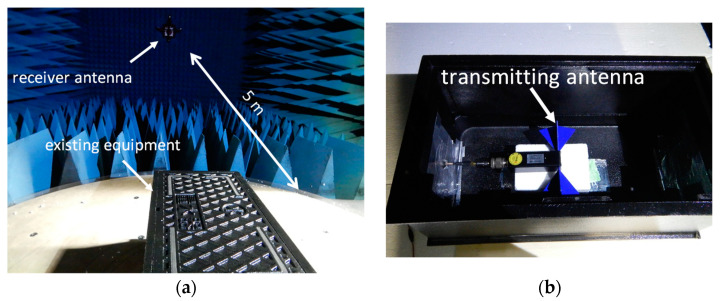
Experimental situation. (**a**) Anechoic chamber measurement situation and (**b**) transmitting antenna.

**Figure 14 sensors-23-09716-f014:**
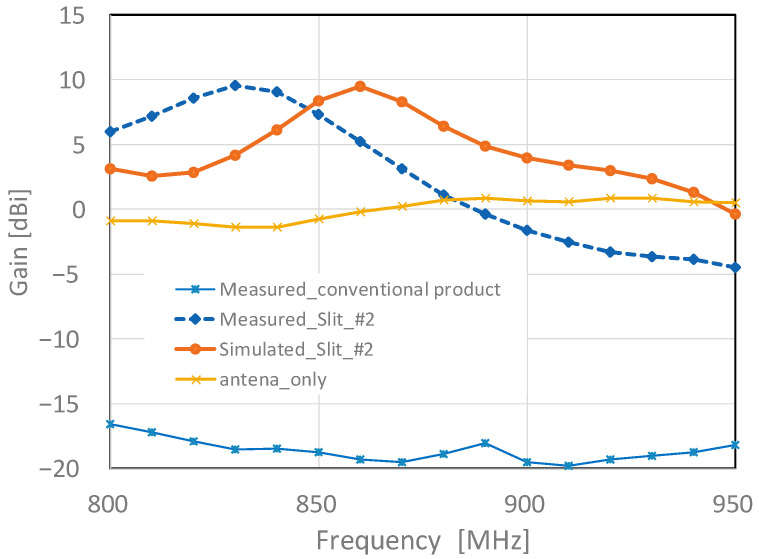
Frequency–gain experimental results (800 to 950 MHz).

**Figure 15 sensors-23-09716-f015:**
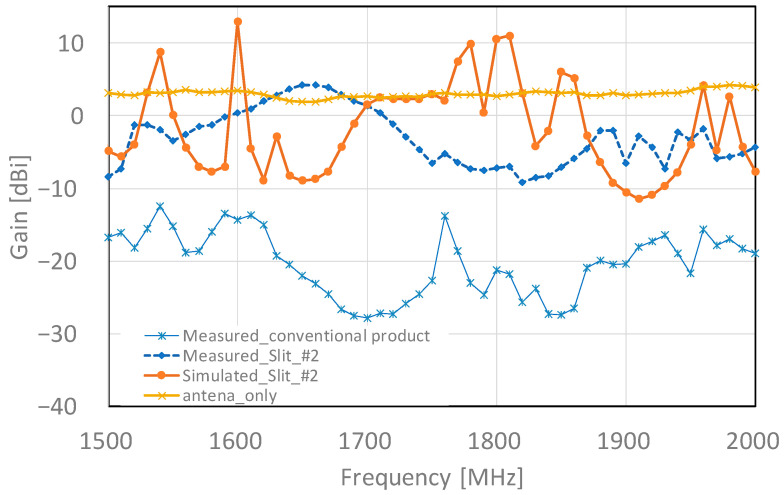
Frequency–gain experimental results (1500 to 2000 MHz).

**Figure 16 sensors-23-09716-f016:**
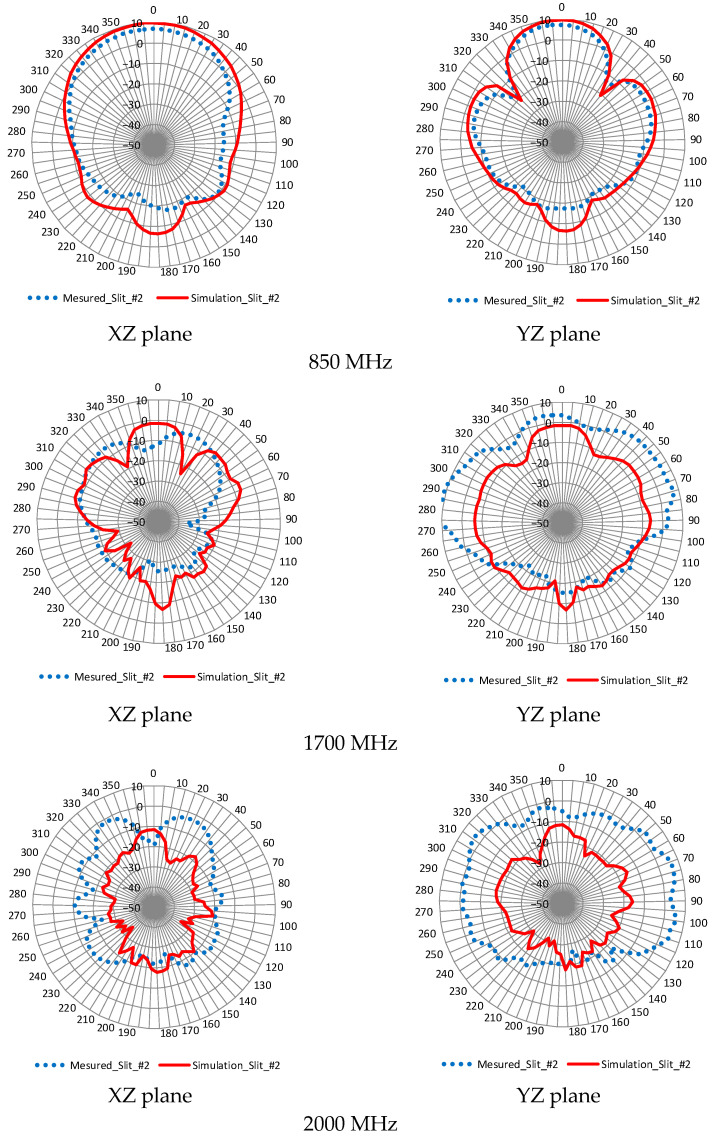
Comparison of experimental and analytical values of radiation patterns.

**Table 1 sensors-23-09716-t001:** Comparison of radio wave transmission structure.

	Radio Wave Transmission Type		Antenna-Mounted Type
	Plastic meter box	Cast iron slit structure	(Manhole antenna)
	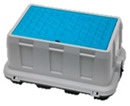	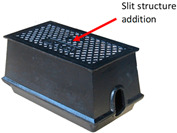	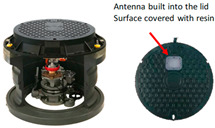
Gain characteristics	−2.0 to 1.5 dBi Transmit antenna specifications (800 MHz–2.5 GHz)	7.35 dBi (850 MHz) 1.58 dBi (1.7 GHz) −4.34 dBi (2.0 MHz)	2.4 to 3.4 dBi (By specific frequencies)
	Good (due to resin)	Best (by slit#2)	Better (built-in antenna)
Applicability Load/Size	Not applicable on driveway	Better	Not applicable in terms of size
Cost	Low cost	Low cost	High cost

## Data Availability

No new data were created or analyzed in this study. Data sharing is not applicable to this article.

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
