# Peer review of "Study on Improvement of Radio Propagation Characteristics of Cast Iron Boxes for Water Smart Meters"

_sensors, 2023, doi:10.3390/s23249716_

Round 1

Reviewer 1 Report

Comments and Suggestions for Authors

The article presents a novel slit structure for cast-iron meter boxes that can improve the wireless communication performance of smart water meters in Japan. The authors claim that their proposed structure can achieve a significant gain improvement in the 800 MHz to 920 MHz band, which is important for the widespread use of smart water meters in Japan. The authors also demonstrate that their structure can satisfy the design loads required for road installation and can be fabricated in a cast iron integral structure. The article is well-written and provides a clear description of the problem, the proposed solution, the experimental setup and the results. The article also discusses the limitations and future work of their study. The article is relevant and timely, as smart water meters are expected to bring many benefits for water utilities and customers in Japan. 

Comments on the Quality of English Language

The quality of the writing and the English in the article is good.

Author Response

Dear Professors:

We would like to express our deepest gratitude for your thorough review and insightful comments on the content of our manuscript.

In the revised version of the paper, all comments and suggestions expressed by the editors and reviewers were considered, answered and addressed. Please find our modifications in the next page.

Reviewer 2 Report

Comments and Suggestions for Authors

The paper is quite well structured and adequately presents the conducted research (justification, analysis, simulations, measurements, comparisons). However, the paper should be improved in order to address the following issues:

1. The Subsection 3.1 is not necessary and should be eliminated. The title of this Subsestion is misleading, as the material presented therein pertains to antenna-theory elements, which are relevant but are not utilized below, as Subsections 3.2 and 3.3 present structure designs modeled/simulated with FEMTET (commercial package exploiting finite elements).

2. The references list contains some works that are tangent to the conducted research (e.g., not all of [19-33] are directly relevant to the antenna problem under study).

3. It would be nice and helpful to provide some details regarding the simulations performed (i.e., modeling attributes, convergence study, computational resourses, execution times).

Comments on the Quality of English Language

Regarding syntax/wording, no major issues were found.

Author Response

(The authors gave the same response as above.)

Reviewer 3 Report

Comments and Suggestions for Authors

The paper presents an interesting study on enhancing smart water meter functionality in Japan through a low-cost slit structure for cast-iron meter boxes. However, there are several limitations and areas that require further clarification or improvement:

1.     The paper does not adequately address how environmental factors (like soil type, moisture levels, or urban infrastructure) might affect the propagation characteristics. Including this analysis would provide a more comprehensive understanding of the technology's effectiveness in different settings.

2.     The long-term durability of the slit structure under various environmental conditions and the maintenance requirements over time need further elaboration. This is particularly important given that these meter boxes are typically buried and exposed to varying conditions.

3.     he authors should analyze the spectral efficiency of the proposed solution, particularly how it manages bandwidth utilization in the specified frequency ranges. This is crucial for understanding its impact on overall network performance, especially in dense urban areas where spectrum allocation can be a significant challenge.

4.     There is a need for a detailed interference analysis, considering both intra-system and inter-system interference. The paper should address how the slit structure design mitigates potential interference with other communication systems operating in proximity, especially given the dense radio frequency environment in urban settings.

5.     The authors should consider the electromagnetic compatibility of the proposed design, including its compliance with international EMC standards. This is vital to ensure that the new design does not inadvertently introduce electromagnetic interference (EMI) that could affect other devices.

6.     discussion on the impact of multipath fading and urban canyon effects on the radio signal propagation in the context of the proposed slit structure is necessary. These factors can significantly affect signal quality in urban environments. Furthermore the strong channel modeling are also need to be discussed, for this purpose the author can following the papers

Channel modeling for UAV-to-ground communications with posture variation and fuselage scattering effect," IEEE Trans. Commun., vol. 71, no. 5, pp. 3103-3116, May 2023.

“A UAV-Aided Real-Time Channel Sounder for Highly Dynamic Non-Stationary A2G Scenarios,” IEEE Trans. Instrum. Meas., vol.72, pp.1-15, Aug. 2023.

7.     The computational model assumes a perfect conductor enclosure, which might not accurately reflect real-world conditions. The impact of these assumptions on the study's findings needs further elaboration.

8.     The study does not appear to consider complex interactions with the surrounding environment, such as soil conductivity, moisture levels, and other factors that can significantly affect electromagnetic wave propagation.

9.     There is a lack of detailed analysis regarding potential interference with other communication systems, especially in urban areas where numerous wireless systems coexist.

10.  More information on the specific properties and composition of the ductile cast iron used and how these properties influence electromagnetic wave propagation would strengthen the study.

11.  While the study explores slit geometry, a more comprehensive analysis of the physical dimensions and shapes of the slits, including their impact on different frequency bands, is needed.

12.  The implications of the slit structure on the manufacturing process and the structural integrity of the meter boxes, especially under various load conditions, are not sufficiently addressed.

13.  Discussion regarding regulatory compliance, especially in terms of electromagnetic radiation safety standards, is missing.

14.  The study should discuss the impact of the proposed design on the maintenance requirements and lifespan of the meter boxes.

15.  The author need to be add the advance wireless communication protocols as discussed in the below article.

Architecture Optimization for Filtered Multicarrier Waveforms in 5GWireless Personal Communications, 2022

Comments on the Quality of English Language

The authors need to be proofread the complete manuscript and remove common mistakes like comma, spelling, and long sentences. 

Author Response

(The authors gave the same response as above.)

Round 2

Reviewer 3 Report

Comments and Suggestions for Authors

The authors have addressed the comments successfully. However, in proofreading the authors need to take care of some minor points like. 
800 MHz to 920 MHz can be written as 800 to 920 MHz, similarly, in the whole manuscript. 

Comments on the Quality of English Language

There are minor grammatical   mistakes in the revised version, like double comma, comma. The authors need to give a look before final submission. 

Author Response

Dear Professors:

We would like to express our deepest gratitude for your thorough review and insightful comments on the content of our manuscript.

In the revised version of the paper, all comments and suggestions expressed by the editors and reviewers were considered, answered and addressed. Please see the attachment.
